# The GIAB genomic stratifications resource for human reference genomes

Nathan Dwarshuis [1], Divya Kalra[2], Jennifer McDaniel [1], Philippe Sanio[3], Pilar Alvarez Jerez [4,5], Bharati Jadhav[6], Wenyu (Eddy) Huang [7], Rajarshi Mondal [8], Ben Busby [9], Nathan D. Olson [1], Fritz J. Sedlazeck [2,7], Justin Wagner[1], Sina Majidian [10,11,12] ✉ & Justin M. Zook [1,12] ✉

Despite the growing variety of sequencing and variant-calling tools, no workflow performs equally well across the entire human genome. Understanding context-dependent performance is critical for enabling researchers, clinicians, and developers to make informed tradeoffs when selecting sequencing hardware and software. Here we describe a set of "stratifications," which are BED files that define distinct contexts throughout the genome. We define these for GRCh37/38 as well as the new T2T-CHM13 reference, adding many new hard-to-sequence regions which are critical for understanding performance as the field progresses. Specifically, we highlight the increase in hard-to-map and GC-rich stratifications in CHM13 relative to the previous references. We then compare the benchmarking performance with each reference and show the performance penalty brought about by these additional difficult regions in CHM13. Additionally, we demonstrate how the stratifications can track context-specific improvements over different platform iterations, using Oxford Nanopore Technologies as an example. The means to generate these stratifications are available as a snakemake pipeline at https://github.com/usnistgov/giab-stratifications. We anticipate this being useful in enabling precise risk-reward calculations when building sequencing pipelines for any of the commonly-used reference genomes.

The last few decades have brought a vast array of increasingly-powerful sequencing platforms and associated software to read DNA molecules. However, no tool or pipeline performs equally across all genomic contexts within the human genome. Particularly difficult genomic contexts include large duplications and large repeats.

Additionally, many sequencing platforms have relatively low performance in homopolymers, and platforms that perform better in homopolymers use short-reads which lack the mapping advantage long reads have in large repeats. The mappers and variant callers used to analyze reads from these platforms also bring context-specific

[1]Material Measurement Laboratory, National Institute of Standards and Technology, Gaithersburg, MD., USA. [2]Human Genome Sequencing Center, Baylor College of Medicine, Houston, TX, USA. [3]University of Applied Sciences Upper Austria - FH Hagenberg, Hagenberg im Mühlkreis, Austria. [4]Center for Alzheimer's and Related Dementias (CARD), National Institute on Aging and National Institute of Neurological Disorders and Stroke, National Institutes of Health, Bethesda, MD 20892, USA. [5]Department of Neurodegenerative Disease, UCL Queen Square Institute of Neurology, University College London, London, UK. [6]Department of Genetics and Genomic Sciences and Mindich Child Health and Development Institute, Icahn School of Medicine at Mount, Hess Center for Science and Medicine, New York, NY, USA. [7]Department of Computer Science, College of Engineering, Rice University, Houston, TX, USA. [8]Department of Bioinformatics, Pondicherry University, Pondicherry, India. [9]DNA Nexus, Mountain View, CA, USA. [10]Department of Computational Biology, University of Lausanne, Lausanne, Switzerland. [11]SIB Swiss Institute of Bioinformatics, Lausanne, Switzerland. [12]These authors contributed equally: Sina Majidian, Justin M. Zook. ✉e-mail: sina.majidian@unil.ch; justin.zook@nist.gov

performance implications due to the assumptions (implicit or explicit) they often make when processing sequencing data[1]. Therefore, improving and fully utilizing the sequencing landscape will require detailed analysis of how different tools perform in a given genomic context.

To this end, we previously developed "genome stratifications" which are carefully-defined browser extensible data (BED) files that divide the human genome into meaningful contexts for benchmarking. The genomic stratifications were originally developed in collaboration with the Global Alliance for Genomics and Health (GA4GH)[2] and are being further developed by the Genome in a Bottle Consortium (GIAB). Coding regions, low mappability regions, high GC content regions, and various types of repetitive regions are examples of such genomic stratifications, and these are currently defined with regard to two linear references, GRCh37 and GRCh38. These stratifications are designed to be used with benchmarks such as those developed by GIAB, which generates variant benchmarks for a set of human genomes to enable development, optimization, evaluation, and comparison of sequencing technologies and variant detection methods[3–5]. GIAB has expanded its variant calling benchmark sets to include increasingly challenging genomic regions and variants as sequencing technologies, variant detection methods (for single nucleotide variants (SNVs), insertions and deletions (INDELs), and structural variants), and assembly algorithms improve[6–8]. As these benchmarks include highly challenging regions, stratifications become increasingly important to understand where methods perform well or have limitations[1,2]. While stratifications are designed for stratifying variant-calling performance when using a benchmarking tool such as hap.py[2] or truvari[9], the stratifications in principle are tool- and application-agnostic.

Genomic stratifications provide value for many users in the genomics community, including bioinformatics method developers, sequencing technology developers, and clinical laboratories. For those developing software tools, stratifications can be used to better understand the advancements or limitations of new methods and identify biases across methods or technologies. For example, stratifications were used in the precisionFDA challenge V2 to evaluate the performance of different technologies across different repetitive regions, such as homopolymers or segmental duplications[1]. Additionally, with respect to evaluating the performance of calling INDELs, this study revealed that the INDEL recall and precision metrics are lower when using PCR amplification compared to PCR-free sequencing calculated for the whole genome. However, these values are almost equal when considering all regions except homopolymers or tandem repeats. This observation demonstrates the importance of genomic stratifications and how they can highlight differences between different technologies and pipelines, allowing for critical investigation[2]. This information can help end users make informed tradeoffs when selecting tools, where performance versus runtime, server cost, hardware requirements, reagent costs, and user expertise must be balanced.

Stratifications are also important in medical practice, both at the research level and in the clinic. For the researcher, stratifications indicate genomic regions where "difficult" variants might be found and as such might require additional resources to study accurately. Stratifications also carry some functional and/or structural information, such as specifying which regions contain coding genes[10] or high GC content, which is useful for designing experiments and association studies. For the clinician, stratifications provide a means to assess confidence in a result. Guidelines for validating clinical pipelines include validating "representative" variants of different types and genome contexts, and stratifications define genome contexts that are challenging[11]. If a patient presents with a pathogenic variant, stratifications can show if this variant resides within a "difficult" region, which in turn could provide a proxy for how much the clinician can trust the

result. Thus, stratifications are instrumental in the development and understanding of variants across different disciplines.

Here we present an update to the previous stratifications, both defining them in terms of the new CHM13 reference as well as exploring novel genome contexts that may be useful stratifications in the future. CHM13 was recently published as the first Telomere-to-Telomere (T2T) reference[12] which completed the remaining 8% of gaps present in the existing references, adding ~2000 genes and ~100 protein coding sequences. In other words, CHM13 provides gapless assemblies for all chromosomes by introducing about 200 Mbp, covering both euchromatic and heterochromatic regions. Furthermore, it includes centromeric satellite arrays, segmental duplications, and the short arms of all five acrocentric chromosomes[12,13]. Overall CHM13 has been shown to improve sequencing data analysis including variant calling[12]. To fully leverage such a reference genome and assess the reliability of existing methods, a set of genomic stratifications for CHM13 is needed. This will facilitate the study of hundreds of new genes and their role in phenotypes or diseases. Moreover, developers of sequencing technology and genome assessment pipelines can benefit from CHM13 and associated stratifications in understanding performance in difficult, newly assembled regions.

The current study also explores new genomic contexts corresponding to different error mechanisms, which may become additional stratifications in the future. For example, it is much easier to call a variant in a tandem repeat if it is the only variant in the repeat; additional variants could "shift" the representation of the variant being called which makes variant calling and variant comparison challenging. Similarly, current stratifications do not account for read coverage or distance between variants. The former is important as higher coverage may imply less difficulty. The latter hinders variant calling where variants are closer together probably due to representational challenges.

## Results

The GIAB stratification resource is a publicly available dataset for the human reference genome. Here we describe the extension of this resource to the CHM13 reference genome (Table 1). We also provide an insight to the differences between three reference genomes: GRCh37, GRCh38 and CHM13. Furthermore, we explore three new features for future stratifications of GRCh38 which include variant complexity in tandem repeats, distribution of genomic distance between consecutive variants, and read coverage of each variant. We now automated generating all stratifications to further facilitate the creation across upcoming complete human genomes or other reference genomes that have been annotated with RefSeq, RepeatMasker, segmental duplications, and Tandem Repeat Finder: https://github.com/usnistgov/giab-stratifications.

### Extending CHM13 stratifications

Coding sequences (CDS) are the regions of the genome that code for proteins which are usually targeted for many clinical tests[6]. Using the CHM13v2.0 assembly and available RefSeq annotations[10], we were able to extract gene coding regions and compare them with those of GRCh38 and GRCh37. The methods used for generating CDS stratifications for GRCh38 were applied to CHM13v2.0[1]. Figure 1 shows the comparison between the total length of the CDS region and its ratio over chromosome length (excluding unknown bases in the assembly which are noted with N) for the all three references, GRCh37, GRCh38 and CHM13. All three reference genomes have similar CDS coverage across chromosomes. Also, the differences that we see in chromosomes 9, 19, and 22 are likely due to the fact that many new bases were added in CHM13 relative to GRCh38 on these chromosomes as they have large centromeres and heterochromatin that were excluded in GRCh38[12].

Reference mappability is a metric that can be used to identify whether reads of a given length will align uniquely to that region of the

**Table 1 | Overview of existing and newly-added stratification types for the three reference genomes**

|  | Stratifications | GRCh37 | GRCh38 | CHM13 |
|---|---|---|---|---|
| Established set | Gene coding regions | ✓ | ✓ | This study |
|  | Functional, technically difficult to sequence | ✓ | ✓ | NA |
|  | GC content | ✓ | ✓ | This study |
|  | Genome specific | ✓ | ✓ | NA |
|  | Regions with low complexity sequences | ✓ | ✓ | This study |
|  | Other difficult genomic regions | ✓ | ✓ | This study |
|  | Segmental duplications | ✓ | ✓ | This study |
|  | Chromosome XY specific regions | ✓ | ✓ | This study |
|  | Patterns of local ancestry | X | ✓ | NA |
|  | Low mappability regions | ✓ | ✓ | This study |
| Exploratory set for future versions | Variant complexity in tandem repeats |  | This study |  |
|  | Distribution of genomic distance between consecutive variants |  | This study |  |
|  | Read coverage of each variant |  | This study |  |

Checkmarks denote existing stratifications. "X" denotes stratifications that were not produced previously for a given reference. "This study" denotes stratifications that were added in this work. Stratifications denoted with NA are not covered in this study.

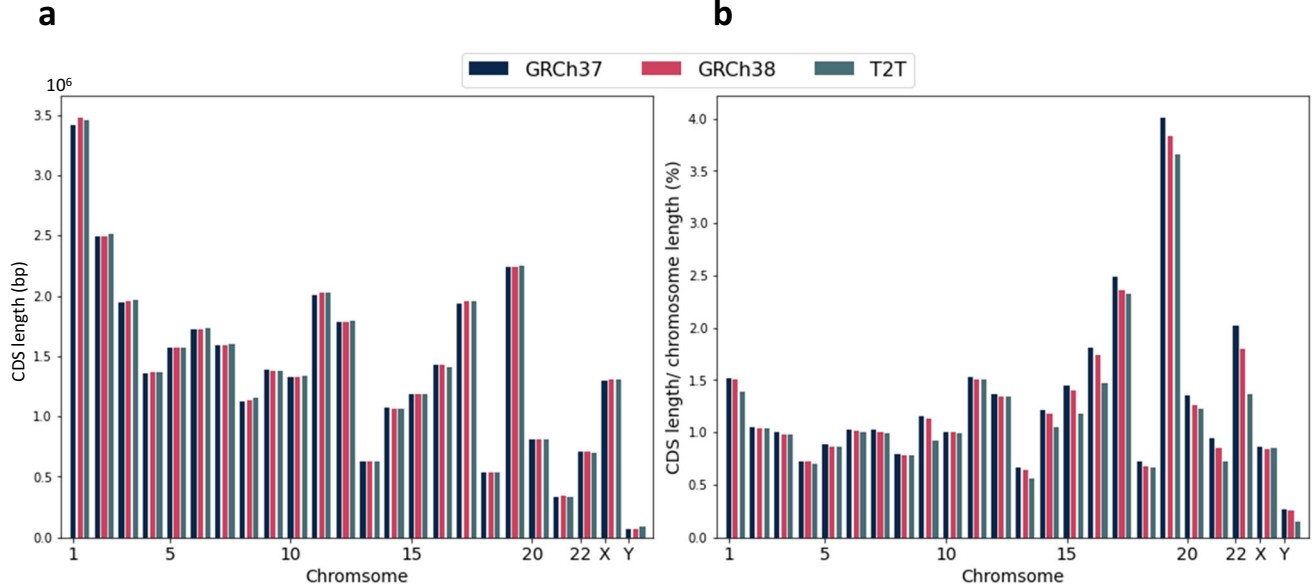

**Fig. 1 | Statistics of the gene coding sequences (CDS). a** Total length of CDS regions for GRCh37, GRCh38 and CHM13. **b** The ratio of length of CDS regions over chromosome length (excluding unknown bases) for GRCh37, GRCh38 and CHM13.

genome (lower means to harder-to-map). The regions that are found to be of low mappability had been previously generated for GRCh37 and GRCh38. We calculated such regions for the CHM13 reference genome at two different stringencies. For moderately low-mappable regions, we permitted up to two mismatches and one INDEL between each 100 bp region and any other region. However, for highly low-mappable regions, we permitted no mismatches or INDELs between each 250 bp region and any other region. Note that 100 and 250 bp correspond approximately to two common lengths used for short-read sequencing, so these regions can be interpreted as those that are difficult to map for short reads. The total length of low mappability regions for each chromosome is depicted in Fig. 2 for all three reference genomes. The total length of the regions in the CHM13 is higher than that of the older references.

To further explore the differences in hard-to-map regions between references, we plotted intra-chromosomal coverage of all low-mappability regions for CHM13 and GRCh38 (*i.e.*, 100 and 250 bp stratifications together, Supplementary Fig. 1). Generally, each

reference had large spikes near the center of each chromosome, corresponding to the centromeres which are known to be repetitive. However, there were some major increases in CHM13 relative to GRCh38. First, we observed large increases in chromosomes 1 and 9; both of these chromosomes have large satellite repeats which would explain this increase. Additionally, we observed large increases in the short arms of chromosomes 13, 14, 15, 21, and 22, which contain large rDNA arrays which are highly repetitive and thus difficult to map[14]. Finally, chromosome Y showed a large increase for over half the chromosome, which can be explained by the large number of ampliconic regions which were added to CHM13[13].

Defining high and low GC-content regions (i.e., the fraction of G and C bases is high or low) is important as different sequencing technologies can produce distinct error profiles in GC-rich and AT-rich regions[15]. This stratification delineates regions with specific amounts of GC content. These are based on the method used for generating standardized GC content BED files by the GA4GH Benchmarking Team and the GIAB. Of note, we consider 10 different ranges from 15 to 85%

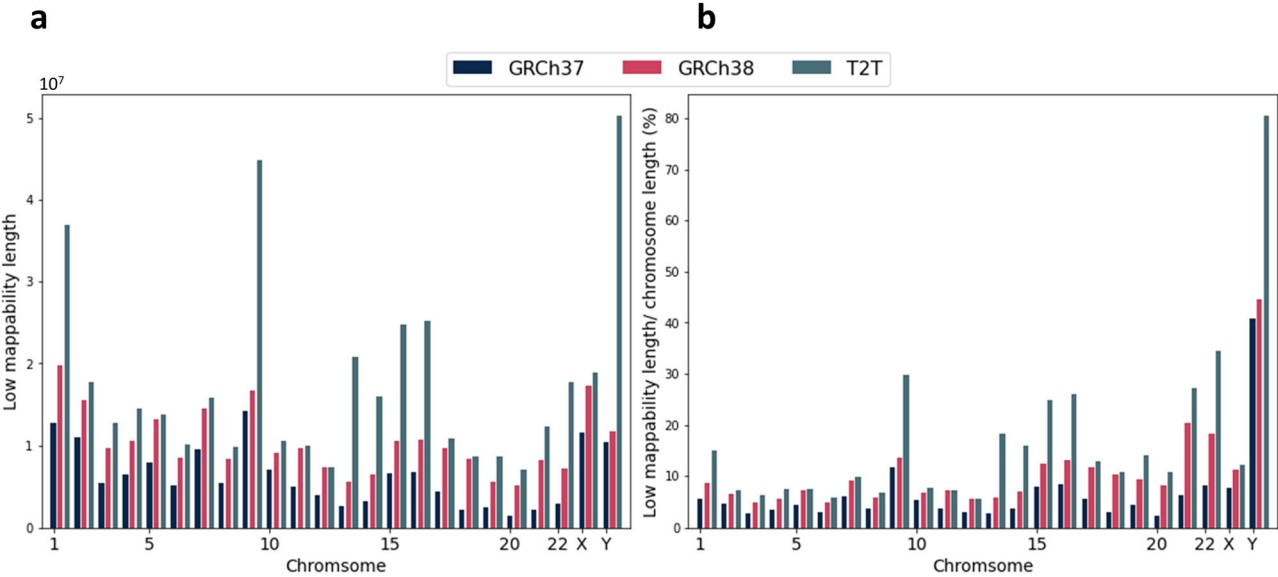

**Fig. 2 | Statistics of low-mappability regions in GRCh37, GRCh38 and CHM13. a** Total length of low-mappable regions, and **b** Ratio of total length over chromosome length (excluding unknown bases) for GRCh37, GRCh38 and CHM13.

of GC content with interval length of 5% in addition to two cases of regions with GC content of smaller than 15% and greater than 85%.

As an example, we depicted the total length of regions for each chromosome of the human reference genomes GRCh37, GRCh38 and CHM13 with the GC content in the range of 20-25% in Fig. 3a. The ratio of total length of regions over chromosome length (excluding unknown bases) is illustrated in Fig. 3b and d. As we can see, three reference genomes follow a similar pattern except for chromosome 13 in CHM13. Moreover, Fig. 3c shows the total length of regions with GC content higher than 85% for GRCh37, GRCh38 and CHM13. Upon investigating intra-chromosomal coverage of regions with >85% GC content, we observed a large increase in the short arms of chromosomes 13, 14, 15, 21, and 22 in CHM13 relative to GRCh38, corresponding to the rDNA arrays (Supplementary Fig. 2).

Three challenging, medically-relevant regions within human genomes including the Major Histocompatibility Complex (MHC), variable/diversity/joining (VDJ) and Killer-cell immunoglobulin-like receptor (KIR) are considered here[16]. These three regions are all highly polymorphic and underpin key immunological functions: the MHC region contains the Human Leukocyte Antigen (HLA) genes which determine "donor matches," the VDJ regions are randomly recombined to produce the T and B cell receptors in T and B cells, respectively, and the KIR region codes for one of the key effector receptors on natural killer cells. The total length of each region on the three reference genomes, GRCh37, GRCh38 and CHM13, are reported in Table 2. As shown, the regions are located on the same chromosome across different reference genomes with comparable total length.

### Evaluating the utility of stratifications for benchmarking

We demonstrated the usefulness of these stratifications for their most common use case which is benchmarking variant caller performance across the genome using a benchmarking tool called hap.py from the GA4GH Benchmarking Team[2].

First, we assessed the differences between the three references within different region types. To do this we utilized our draft assembly-based benchmark for HG002, which was constructed from the HG002 T2T Q100 v1.0 diploid assembly (see Methods)[13]. This benchmark is made from a complete, accurate assembly as opposed to the current mapping-based callsets from GIAB. Thus, it includes more difficult regions than were previously available and can be created from the

alignments of the assembly to any reference, which makes it well-suited to comprehensively test these three references. We then benchmarked a HiFi-Deepvariant callset (as the query) using the draft assembly-based benchmark (as the truth) and evaluated precision and recall of variants (Fig. 4A). In this figure, the value of each bar is the aggregated Phred-scaled score, and the error bars are the estimated 95% binomial confidence intervals.

Across many stratification categories, CHM13 had a lower score than GRCh38, which in turn also had a lower score than GRCh37. This can be explained by the fact that each new reference progressively added more difficult regions to the human genome as technology improved with time. The largest differences between references were for SNVs in segmental duplications and low-mappability regions, which have been increasingly included in GRCh38 and CHM13. CHM13 corrected some false segmental duplications in GRCh38 and added segmental duplications missing in GRCh38 and GRCh37[17], which can cause lower accuracy in GRCh38 and GRCh37. However, these callsets used the GIABv3 refined version of GRCh38 that masks the false duplications and adds decoys for a few missing sequences[18], as well as the version of GRCh37 that includes the hs37d5 decoy. In addition, the overall SNV precision and recall for TRs and HomoPolymer (HPs) was lower than some other stratification types such as low-mappability regions (lowmap) and high/low GC. This is likely due to the HiFi platform used to generate the callset, which is known to be more error-prone in these repeat regions[1]. However, when focusing on our stratification excluding difficult regions, performance was similar across references. Furthermore, the scores for INDELs were lower than SNVs across all stratifications and metrics, which tend to be more challenging than SNVs due to alignment challenges. INDELs also likely bias to lower scores due to hap.py using vcfeval[19] for comparing variants, which does not give partial credit for complex INDEL variants that frequently occur in TRs and HPs[20]. In summary, this shows how stratifications can be used to understand how performance differs across the three references. It also justifies the use of CHM13 over the other previous references, as using GRCh38/37 could result in an inflated score depending on the regions under study.

Next, we asked how the new regions (i.e., "nonsyntenic") in CHM13 relative to GRCh38 specifically impacted performance. The nonsyntenic regions contain 1000 s of variants (Supplementary Fig. 3), many of which probably have biological significance but have been

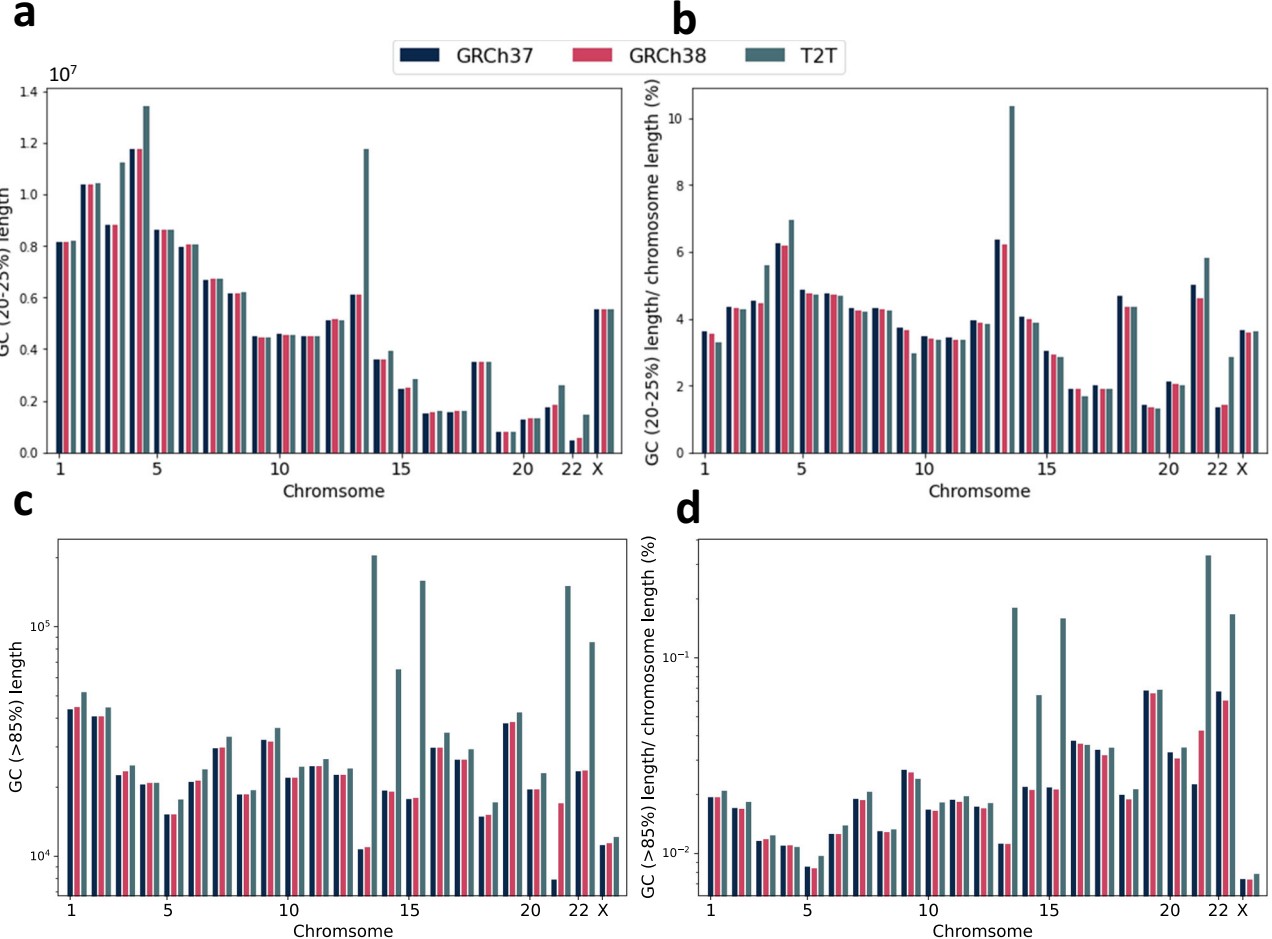

**Fig. 3 | Statistics of regions with specific GC content for GRCh37, GRCh38 and CHM13. a** Total length of regions with GC content in the range of 20–25%. **b** Ratio of total length of regions with 20-25% GC content over chromosome length (excluding unknown bases). **c** Length of regions with GC content higher than 85% in log scale. **d** Ratio of total length of regions with GC content >85% over chromosome length in log scale (excluding unknown bases).

difficult to study without a complete reference such as CHM13. We modified the analysis in Fig. 4A to subset to either the syntenic or nonsyntenic regions prior to benchmarking using the targeted flag in hap.py (see Methods) (Fig. 4B). 93% of the benchmark variants in nonsyntenic regions were in our stratification that contained all difficult regions. For INDELs and SNVs, the difference in Phred score between syntenic and nonsyntenic regions were about 5 and 15 respectively for both precision and recall across all stratifications (nonsyntenic being much lower in general). In summary, this indicated that the changes from GRCh38 to CHM13 indeed included much more difficult regions, and our stratifications enable understanding of the differences between performance metrics across references.

Additionally, we used the stratifications to assess the performance improvements of recently published software components in the Oxford Nanopore Technologies (ONT) variant calling pipeline, specifically guppy and clair, which are a base caller and variant caller

respectively developed specifically for ONT reads (Fig. 4C)[21]. The ONT reads were acquired from HG003, and the same reads were used for both versions of guppy/clair (guppy4+clair1 and guppy5+clair5). The experimental setup was similar to that of Fig. 4A and B except that we used the v4.2.1 GIAB HG003 benchmark since the GIAB team has yet to build an assembly-based benchmark for HG003. We observed that in general, there was a substantial performance gain (up to 10 Phred-scaled) between the old and new caller versions, as expected. However, this gain was not uniform. For HPs and/or TRs, the precision/recall metrics were less than that for "Autosomes" (representing a global mean for all autosomes), which in turn was less than stratifications which excluded HPs and TRs. For SNVs in particular, HPs and TRs had modest performance gain for precision but substantial gain for recall. This agrees with previous results that the ONT platform has a higher error rate in homopolymers[1]. In totality, this shows how the stratifications can be used to benchmark performance of new technologies as they evolve, which can serve as a valuable resource both for those developing these platforms as well as consumers who need to buy or update their workflows.

Finally, we used the draft Q100 benchmark with our new stratifications to compare performance metrics for variants called from HiFi and Illumina reads aligned to the CHM13 reference genome and called with DeepVariant (Fig. 4D). While performance in many regions is similar between these HiFi and Illumina DeepVariant callsets, the stratifications highlight a few notable differences. For example, regions with low-mappability (defined for short reads) and segmental

**Table 2 | Total length of three difficult genomic regions, namely MHC, VDJ and KIR, on reference genomes GRCh37, GRCh38 and CHM13**

|      | Chromosome(s)   | GRCh37    | GRCh38    | CHM13     |
| ---- | --------------- | --------- | --------- | --------- |
| MHC  | 6               | 4,970,558 | 4,970,558 | 4,920,493 |
| VDJ  | 2, 7, 14, and 22 | 3,232,649 | 3,348,726 | 3,541,519 |
| KIR  | 19              | 155,001   | 155,044   | 156,419   |

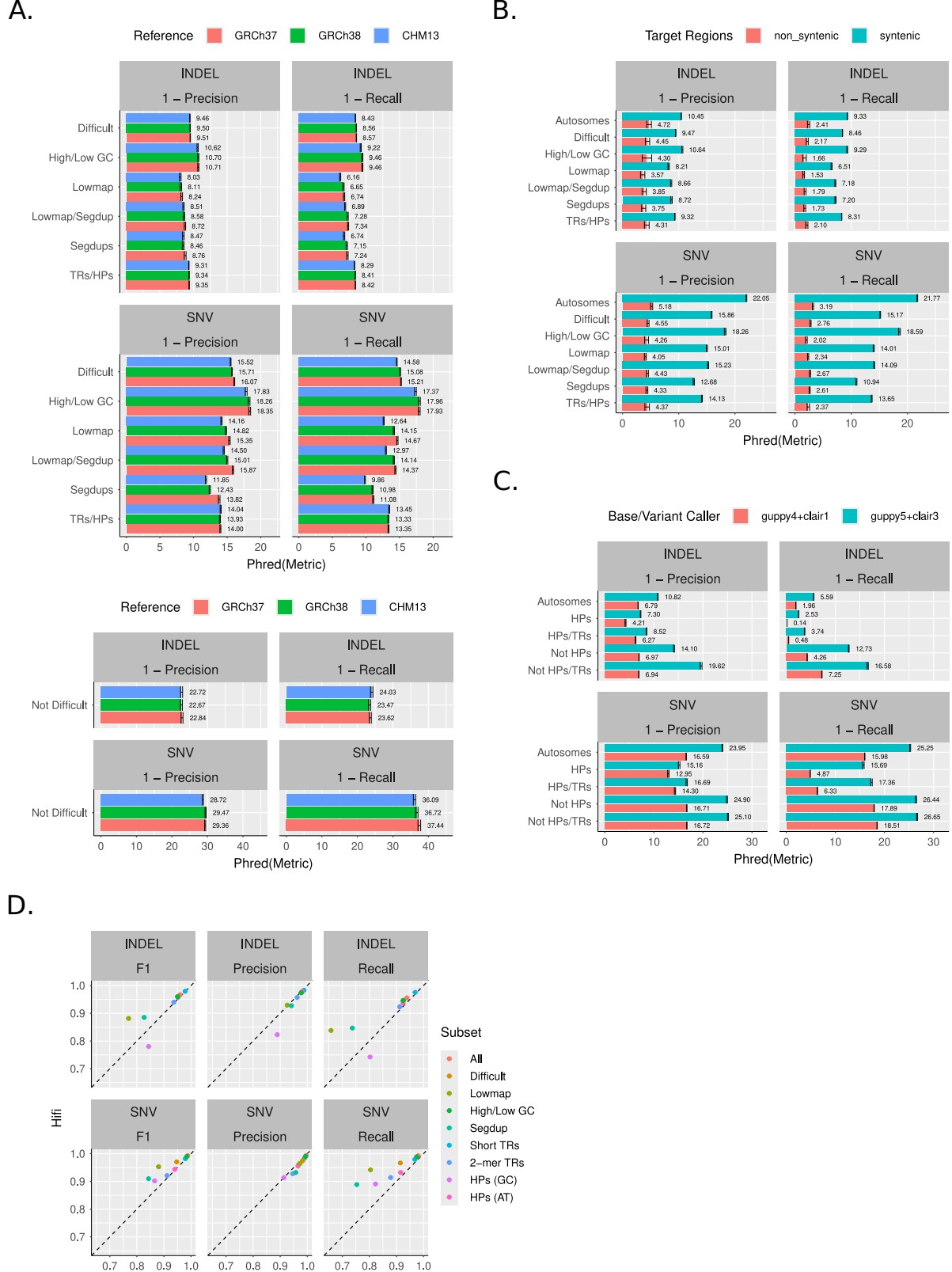

duplications experienced lower recall of both SNVs and INDELs for Illumina-DeepVariant. Additionally, recall in low-mappability and segmental duplication regions for HiFi-DeepVariant was lower than the other stratifications at about 85% for INDELS and 90-95% for SNVs, indicating that there is still room for improvement even with long reads. Because CHM13 includes more low-mappability regions and

segmental duplications, our stratifications for CHM13 are particularly important. In addition, our stratifications for GC vs. AT homopolymers help highlight how AT homopolymers are similar between the callsets, whereas HiFi-DeepVariant had higher accuracy for SNVs in GC homopolymers and Illumina-DeepVariant had higher accuracy for INDELs in GC homopolymers. Importantly, our stratifications help the

**Fig. 4 | Stratifications reveal nuances in precision and recall performance when benchmarking using hap.py. A** Performance within important stratifications using assembly-based HG002 benchmark and GRCh37, GRCh38, or CHM13 as reference and a HiFi-DeepVariant query callset. **B** The CHM13 performance results from (**A**) compared with the same benchmarking pipeline restricted to nonsyntenic regions relative to GRCh38. **C** Performance within all autosomes or tandem repeats/homopolymer regions for ONT callsets created with either guppy4+clair1 or guppy5+clair3. **D** Comparison of HiFi and Illumina callsets on CHM13 using the Q100

benchmark. Each bar is the mean of the given metric which is also shown as text. Error bars are 95% binomial confidence intervals computed with the Wilson method (see Methods). Stratification meaning on y axes: Lowmap = low-mappability regions (100 and 250 bp sizes); High/Low GC = GC content > 25% or > 65%; SegDup = segmental duplications >= 1 kb; TRs = tandem repeats; HPs = homopolymers >= 7 bp or imperfect homopolymers >= 11 bp; Difficult = SegDup+LowMap+HPs+TRs +XY PAR/XTR/Ampliconic+High/Low GC; Autosomes = all autosomal regions; 2-mer TRs = repeats with unit size 2; Short TRs = tandem repeats <50 bp long.

community continue to evaluate the performance of long and short read sequencing technologies in different genomic contexts as new technologies and analysis methods are developed.

## Exploring new features for future stratifications

In this study we expanded many of the stratifications to CHM13. We also explored possible three new stratifications for GRCh38 in addition to the so far described well-established stratifications (Table 1). We examined variant complexity in tandem repeats, distribution of genomic distance between consecutive variants, and read coverage of each variant.

Tandem repeats frequently contain complex variants (*e.g.*, multiple SNVs and INDELs), which can cause errors in variant calling and benchmarking[22]. Variant callers tend to produce more errors in repeats because variants could be mis-identified among the repeat, particularly if reads are insufficiently long to contain the entire repeat. Notably, such errors are likely to increase when there is more than one variant in the region, because if any variants are filtered incorrectly then all variants in the region can be counted as FPs and FNs due to differences in representation[20]. It may thus be sensible to create new stratifications for tandem repeats categories by number of variants, where number of variants corresponds to difficulty.

To understand how variants were distributed in tandem repeats, we intersected the Q100 variant benchmark of the HG002 sample[23] with the GIAB tandem repeat and homopolymer stratification BED files. This variant call format (VCF) file includes both small variants and structural variants (SVs) (except for inversions and translocations), so that we can assess the full spectrum of variants in tandem repeats[4]. After splitting multiallelic variants, we additionally filtered any variants that overlapped repeat boundaries (~1500 variants), though these would be important sources of complexity to explore in the future. We found that the vast majority (~90%) of the repeats in GRCh38 did not have any variation, but >10,000 tandem repeats contain more than one variant and >1,000 contain more than three variants, resulting in complex variants that can cause challenges in variant calling and variant representation (Fig. 5a).

We further investigated the distributions of repeat region size and variants inside repeats with only one variant. About 30% of such variants were SNVs, and about 50% were INDELs between 1 bp and 2 bps (Fig. 5b). There were several hundred SVs associated with these repeats as well, indicating that some SVs in repeat regions exist without any smaller variants in the same repeat. This distribution was mostly consistent across chromosomes. Furthermore, we investigated the size distribution of repeats with a single variant according to variant type (Fig. 5c). We found that small INDELs are disproportionately more likely to be in repeats <80 bp long compared to other variant types. Of note, the variant density in tandem repeats is more than four times of that in segmental duplications (Fig. 5d, e).

We also assessed the quality of variant calling when variants are in tandem repeat regions. To do so, we considered the Illumina HiSeq X DeepVariant callset as the query and used the HG002 Q100 small variant benchmark as the truth. We stratified the repeat regions based on the number of variants inside. Overall, the results show that more variants in a tandem repeat region corresponds to lower precision and recall in variant calling. In fact, false positive and false negative rates for both SNVs and INDELs are more than 10 times higher in tandem repeats with >10 variants relative to those with a single variant (Fig. 5f).

Tandem repeats with a single variant have lower error rates than variants outside tandem repeats on average.

*Next, we explored the distribution of genomic distance between consecutive variants.* Understanding variant distribution is important, since the likelihood of representational difficulties increases as the distance between any two variants decreases. Furthermore, in extreme cases, too many variants within a region can lead to a reference allele bias due to e.g., mapping biases[24,25]. In addition, high variant density can result from tandem repeats and gene conversions, which can cause mapping errors[26]. Conversely, read-based phasing of variants becomes more challenging as distance between heterozygous variants increases[27–29], which can cause diploid assembly errors[6].

To calculate distance between variants, we again use the draft GIAB benchmark based on the HG002 Q100 diploid assembly aligned to GRCh38. Supplementary Fig. 4 depicts the distribution of the genomic distance between any two consecutive variants for all autosomal chromosomes. This draft benchmark is a relatively comprehensive characterization of variants, though variant density can depend on how variants are represented. To assess how accuracy depends on distance between variants, we analyzed variants called with DeepVariant for HG002 sequenced with Illumina HiSeq X platform (Fig. 5g).

We observed that when variants are close to each other (within 1-10 bp of another variant), the quality of called variants is lower than those variants that are further apart (100–1000 bp), with SNVs having the highest precision (25.5 Phred) with recall (19.8 Phred). Of note, 3.4 million (49.3%) of the 6.9 million variants have a neighboring variant within 100 bp. Interestingly, around one-fifth of the variants are in repetitive regions including tandem repeats and segmental duplications, which may cause variant calling and representation challenges. However, a very high distance (>10kbp) between variants corresponds to lower quality called variants, especially lower precision. This lower precision in part appears to result from the lower density of true variants in the region, decreasing the denominator in the precision calculation, and many false positives are in clusters due to mis-mapping in segmental duplications.

In addition to stratifications mentioned above, here we considered HG002 to explore the read coverage of each variant, where abnormal coverage implies that the region flanking the variant is a difficult region to align or call variants due to potential mapping challenges and/or misrepresentations (e.g., copy number variants). Additionally, the idea behind calculating coverage values is to look for new informative metrics, which can be fed as an additional feature into machine learning models to predict the quality of called variants[30]. Obviously, one would expect sufficient coverage leads to higher variant calling accuracy. However, too high coverage may indicate mismapped reads due to duplications in the individual, which can result in false variant calls. Another cause of abnormal coverage is small deletions and insertions, so we have chosen to look at two different variant types: INDELs and SNVs.

Accordingly, we calculated the average read coverage of genomic positions for each variant in an Illumina-DeepVariant callset with 40x mean coverage (see Methods). As expected, precision and recall were highest near 40x coverage. Variants in regions with low coverage (<20×) had 1–2 orders of magnitude lower precision and recall, likely due to mapping and genotype errors and filtering of true

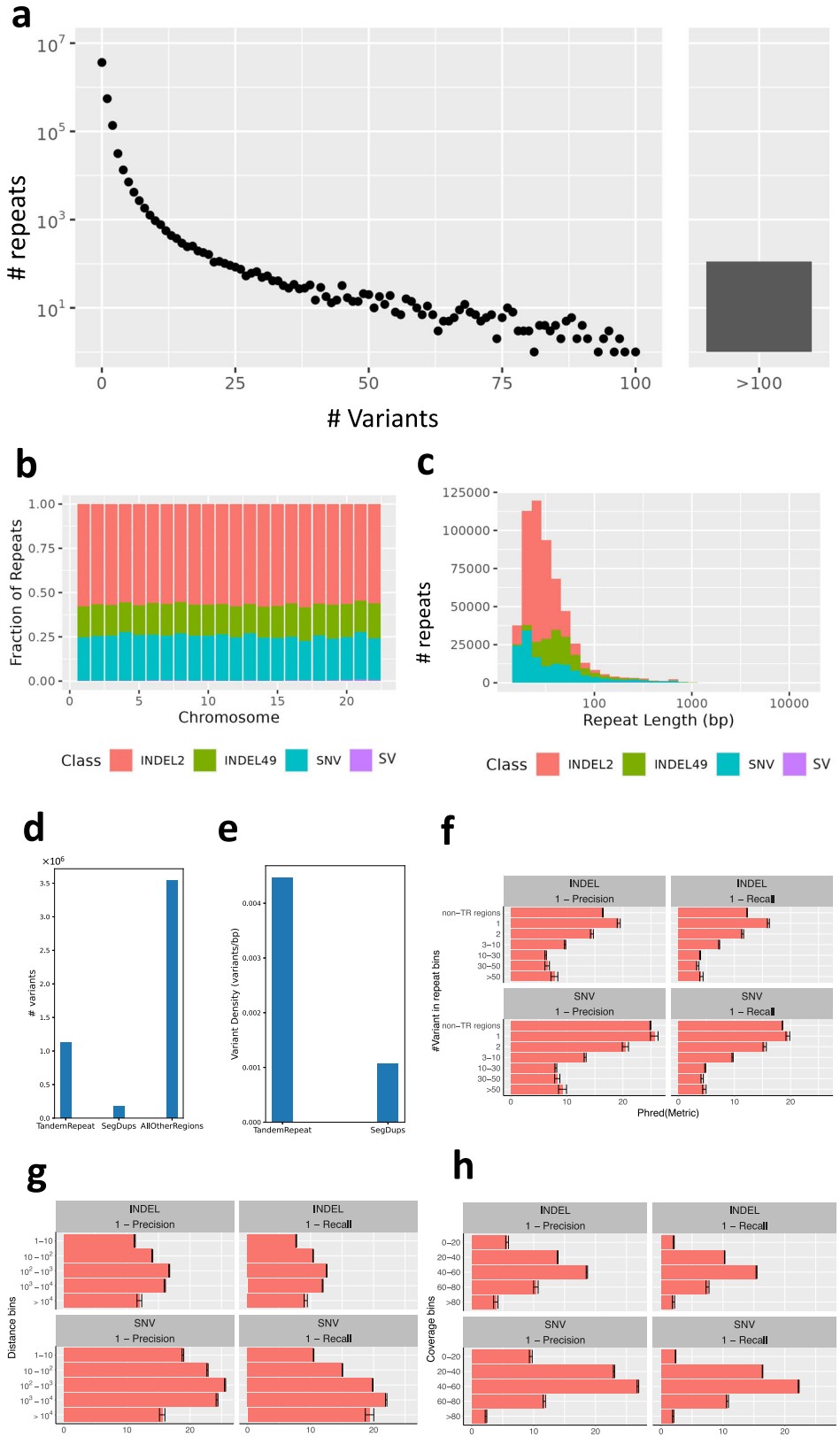

variants (Fig. 5h). Very high coverage >80 also resulted in higher error rates, likely due to mapping errors resulting from duplications in HG002 and/or in GRCh38. These show that variants are most accurate when coverage is near the mean of 40× (i.e., between 20× and 60×), with a Phred score larger than 20, equivalent to precision/recall > 0.99 (Fig. 5h).

## Discussion

In this work, we present a complete set of genomic stratifications across GRCh37, GRCh38 and CHM13. We highlight how these unique stratifications are broadly important for deeply understanding accuracy of sequencing and analysis methods, which are continually being improved by the genomics community. These stratifications are

**Fig. 5 | Distribution of SNV and small INDEL variants within tandem repeats throughout GRCh38 using the HG002 Q100 variant benchmark. a** The distribution of the number of variants per repeat. Y-axis shows the number of tandem repeats and x-axis is the number of variants in each tandem repeat. **b**, **c** Among repeats with only one variant, the fraction of the variant class by chromosome **b** and the distribution of intersecting variants classified by type according to repeat length (**c**) INDEL2; INDELs with length <= 2, INDEL49; INDELs with length > 2 and length <= 49, SNV; single nucleotide variants, SV; structural variants. **d** Number of variants in tandem repeats, segmental duplications and all other regions. **e** Variant density in regions of tandem repeats and segmental duplications. **f** Performance within new stratifications using HG002 Q100 benchmark and an Illumina Deep-Variant query callset for tandem repeat regions with different number of variants inside. **g** Performance within regions with different genomic distance between variants. **h** Performance within regions with different coverage values for a variant set called from a BAM file with mean coverage of 40×. For (**f**–**h**) each bar represents the mean of the given metric. Error bars are 95% binomial confidence intervals computed with the Wilson method (see Methods).

designed to be used with benchmarks like those from GIAB, so that benchmarking tools (e.g., hap.py) can output performance metrics for a variety of challenging genomic regions. This enables users to optimize sequencing and analysis methods, or select the best method for a particular application. In addition to benchmarking methods, these stratifications can be leveraged for a variety of other applications in genomics.

For example, GIAB has used these stratifications to exclude problematic regions from particular technologies or from all technologies when defining benchmark regions. In previous benchmarks, variants from long reads were trusted more than short reads in difficult-to-map regions, whereas variants from PCR-free short reads were trusted more in homopolymers. For a new assembly-based benchmark for chromosomes X and Y, long homopolymers are excluded because the assembly was found to be less reliable in these regions[31]. In addition, segmental duplications and satellite stratification regions were excluded from the benchmark region if any breaks occurred in these regions in the assembly to reference alignment. In these ways, robust stratifications are critical for defining these widely-used benchmarks, and similar stratifications for the CHM13 reference are important for ongoing work creating benchmarks on this new reference.

This resource will be beneficial for other applications outside of variant benchmarking. First, the stratifications of regions with low complexity sequences could be used to filter variants in repeat regions[32] and abnormal coverage values could be utilized for quality control of haplotype phasings[33]. Second, the GC content and repeat stratifications could be utilized for assessing the quality of genome assemblies[34,35]. Third, stratifications can be used to systematically assess the sequencing biases in different genomic contexts[36,37]. Such features also have the potential to be used for improving and filtering gene annotations[38], gene expression[39], genome-wide association studies[40]. Fourth, stratifying variant callsets by region can complement the variant quality scores provided by the caller itself, which is often based on a more localized and limited set of data such as read depth and read quality. Finally, our pipeline has enabled stratifications to be generated for additional assemblies, such as the T2T diploid assembly for HG002[41], which can be used to flag difficult regions, which in turn will enable easier constructure of future human diploid assemblies[23,42].

Providing new stratifications for CHM13 is also a valuable addition for the community. Compared to GRCh38 (which is only 92% complete), CHM13 has no gaps (100% complete) and therefore has many new regions that potentially have biological significance, particularly in segmental duplications and centromeres. We showed using our HG002 Q100 benchmark that the nonsyntenic regions include thousands of new variants (Supplementary Fig 4), which can now be studied if one uses CHM13 instead of GRCh37/38 as the reference. Furthermore, moving to a complete reference such as CHM13 raises the standard for performance required to accurately call variants. When comparing short and long reads using our HG002 Q100 benchmark with variants called on CHM13 (Fig. 4D) we found that long reads excelled over short reads in low-mappability and segmental duplications (which comprises most of the nonsyntenic regions). However, these scores (particularly recall) were still lower than other regions such as homopolymers and GC-rich regions, indicating that even with long reads there are still improvements to be made. Thus, using CHM13 as a reference (as well as our corresponding stratifications for it) provides a measuring stick by which future progress in calling these particularly difficult variants in different classes of challenging regions can be assessed.

While previous iterations of these stratifications (https://github.com/genome-in-a-bottle/genome-stratifications) have been partially used in past work[1,2], this manuscript provides a holistic overview and analysis of this resource. In addition to adding CHM13 and several new stratification categories (Table 1), all prior code was unified into a fully automated, reproducible Snakemake pipeline based on publicly-accessible resources which are hashed to verify integrity. This level of transparency and rigor allows this resource to be more trustworthy and accurate for end users. Since previous iterations of the stratifications were generated in a semi-automated fashion, using this unified pipeline also helped us fix several bugs and inconsistencies between each reference, as well as improve naming conventions. We also expanded on some of the previous stratification categories. For example, we added homopolymer stratifications subset to A/T or G/C regions since these pose distinct challenges for many sequencing platforms. Finally, some stratifications were improved by using newer source data, such as SEDEF in the case of segmental duplications (which previously used the superdups track from UCSC). A summary of all changes throughout the development history of this resource can be found in the CHANGELOG.md file at https://github.com/usnistgov/giab-stratifications.

Having a unified snakemake pipeline also allows other users to generate their own stratifications on a reference of their choice, assuming certain conditions are met. First, this was designed for and tested on human haploid references (with diploid in development), but theoretically any haploid (and eventually diploid) reference for any species should work without issue. Second, generating certain stratifications requires external data sources (RepeatMasker and TandemRepeatFinder for example). Some stratifications such as many of the homopolymer files and mappability files only require a FASTA file as input. These requirements are described in Supplementary Table 2. Also, the pipeline can be configured to run with no/partial external data sources, in which case it will generate what it can with the available data. Thirdly, the pipeline currently assumes an X and Y chromosome in the case of sex-specific stratifications, which is appropriate for most applications that rely on mapping to references like GRCh37, GRCh38, and T2T-CHM13v2.0 that include both X and Y chromosomes. XX karyotype compatibility is in development to enable generating stratifications for XX diploid assemblies. Finally, most of the pipeline does not require much memory as many of the steps are stream-based. However, there are hundreds of steps to generate all stratifications for one reference, so having many cores will be beneficial. With 16 CPU cores at 3 GHz, the full pipeline will generally take 12 hours for one reference. A few steps are also memory intensive; generating mappability stratifications will require 16 G of RAM, and running hap.py to test the stratifications in a benchmark scenario will require 48 G of RAM (both for human haploid references). Installing the pipeline should only require Conda or Mamba after the repository is cloned.

In addition to previously used stratifications, we now introduce several new ones that should yield even further insights into different methods. Understanding variant density in tandem repeats will enable one to flag potential representation issues, as multiple variants in a repeat often have multiple valid representations depending on alignment parameters. The variant distance stratifications reflect the complexity of regions enriched for variants with several possible representations, which are generally difficult for software tools to reliably interpret (especially if they are also in repeats). The variant coverage stratification will provide deeper insights into biases due to too high or too low coverage regions (*e.g.*, due to repetitive regions or reduced sequencing performance).

GIAB continues maintaining the current stratifications and generating new ones. Specifically, additional stratifications could be generated for additional pangenome references in the future, such as those being developed by the Human Pangenome Reference Consortium. This work represents a rosetta stone to better understand variant analysis and will be utilized across consortia and single sample projects that rely on standardized stratifications to filter and optimize their methodologies.

## Methods

### Exploring mappability of CHM13 using GEM
We used GEnome Multitool (GEM)-Mapper[43] (version pre-release 3) on the CHM13v2.0 reference genome to create BED files of low mappable regions. We followed available scripts https://github.com/genome-in-a-bottle/genome-stratifications/tree/master/GRCh38/mappability. Briefly, we generated raw mappability files under two stringency levels: low stringency (100 bp single-end reads, two mismatches, and one INDEL) and high stringency (250 bp single-end reads, 0 mismatches, and 0 INDELs). These mappability BED files were then processed with SAMtools and BEDtools to find the union of the two stringency levels for the final BED file with low mappability regions for CHM13v2.0.

After running both mappability scripts, we had four final BED files containing nonuniquely mapped regions for each stringency level, as well as a final BED file containing all low mappability regions when performing the union for both stringencies.

### Generate CDS regions for CHM13v2.0
We used the R script provided for GRCh38 https://github.com/genome-in-a-bottle/genome-stratifications/blob/master/GRCh38/FunctionalRegions/create_GRCh38_cds_bed.Rmd and ported it over to identify the gene coding regions (CDS) in the CHM13v2.0 assembly. It requires R packages - rmarkdown, tinytex, knitr, tidyverse, devtools.

We used the following files from the NCBI FTP site:
- FTBL: https://ftp.ncbi.nlm.nih.gov//genomes/refseq/vertebrate_mammalian/Homo_sapiens/all_assembly_versions/GCF_009914755.1_T2T-CHM13v2.0/GCF_009914755.1_T2T-CHM13v2.0_feature_table.txt.gz
- GFF: https://ftp.ncbi.nlm.nih.gov/genomes/refseq/vertebrate_mammalian/Homo_sapiens/all_assembly_versions/GCF_009914755.1_T2T-CHM13v2.0/GCF_009914755.1_T2T-CHM13v2.0_genomic.gff.gz

Additionally, the script required a .fai index file which was created from the CHM13v2.0 reference assembly.

### Generating GC content BED files using seqtk for CHM13v2.0
We use an existing script created to generate the GRCh38 GC Content Stratification BED files. The script required seqtk version-1.3-r106 tool, bedtools v2.27.1, and tabix v1.9. Three essential data files were required to run the script file: the CHM13v2.0 FASTA, the CHM13 genome file. The genome was converted to BED format by adding a middle column of 0 (such that each line had the length of the entire chromosome). We ran seqtk for various fractions of GC content, all within windows of 100 bp. After running seqtk, we added 50 bp slop to each BED file and merged.

### Lift-over for OtherDifficult regions
In order to find the coordinate of well-studied genes including MHC, KIR, and VDJ that are considered as difficult regions, we performed liftover for such regions from GRCh38 to CHM13v2.0. To obtain the OtherDifficult regions data of the GRCh38 we referred to the reference sample released by the GIAB https://ftp-trace.ncbi.nlm.nih.gov/ReferenceSamples/giab/release/genome-stratifications/v3.1/GRCh38/OtherDifficult/. To perform the lift-over, we used the minimap2 (v2.24) aligner with arguments -ax asm5 followed by bedtools bamtobed and merge (v2.30.0). The resulting BED files are provided as part of the GIAB stratification resource.

### Snakemake pipeline
**Overview.** This work (first done as part of a hackathon) was incorporated into a snakemake pipeline which can be found at https://github.com/usnistgov/giab-stratifications-pipeline and https://github.com/usnistgov/giab-stratifications. The latter repository holds the global configuration for the three references in this work, and references the former repository as a submodule. The former repository is reference-agnostic and encodes the build rules for the stratification files themselves.

For the identity of every input file used to make these stratifications (including hashes), refer to https://github.com/usnistgov/giab-stratifications/blob/master/config/all.yml.

**Stratification validation.** Each stratification file prior to publishing was ensured to meet the following criteria:
- Only contained valid chromosomes (*i.e.*, 1-22, X, Y).
- File was bgzip compressed.
- File was a valid BED file (three columns, tab-delimited, with 2nd and 3rd columns as non-negative integers with 3rd greater than 2nd).
- All regions in the BED file were sorted in numeric order (*i.e.*, chromosomes ordered 1-22, X, then Y with each region then sorted by start and end).
- No regions overlapped with each other.
- No region overlapped a gap region (which included the PAR on chromosome Y)
- No region fell outside chromosomal boundaries.

### Evaluating the utility of stratifications for benchmarking
We created an assembly-based benchmark from the Q100 assembly for HG002. Specifically, the HG002 Q100 small variant benchmark was created using v0.011 of DeFrABB (https://github.com/usnistgov/giab-defrabb), the T2T-HG002-Q100v1.0 diploid assembly (https://github.com/marbl/hg002), and GRCh38 reference (https://ftp-trace.ncbi.nlm.nih.gov/ReferenceSamples/giab/data/AshkenazimTrio/analysis/NIST_HG002_DraftBenchmark_defrabbV0.011-20230725/).

DeFrABB (Development Framework for Assembly-Based Benchmarks) is a snakemake-based pipeline created to facilitate the iterative development of benchmarks sets for evaluating variant callsets using high-quality diploid assemblies (https://github.com/usnistgov/defrabb). DeFrABB first generates assembly-based variant calls using dipcall v0.3 (https://github.com/lh3/dipcall)[44]. Dipcall was run with default parameters with the following Z-drop parameter, -z200000,10000,200, which yielded more contiguous assembly-assembly alignments compared to the default value. After reformatting and annotation, the variant set reported by dipcall (VCF) was used as the draft benchmark variants. Note that we call these "draft" variants since this benchmark has not been officially evaluated and released by GIAB yet; however, GIAB and the Telomere to Telomere Consortium

have polished and curated the assembly and variant calls sufficiently for it to be used for this analysis.

The benchmark regions (analogous to the "confident regions" in the GIAB v4.2.1 small variant benchmarks) are defined as regions with a 1:1 alignment between each assembled haplotype and the reference (except chromosomes X and Y). These regions excluded gaps in the assembly and their flanking sequences, as well as any large repeats (satellites, tandem repeats >10 kb, and segdups) that have a break in the assembly to reference alignment on either haplotype. Additionally structural variants including repeat regions where SVs overlapping large tandem repeats are also excluded from the benchmark regions. Widened SV coordinates were identified using the SVanalyzer v0.36 widen module (https://github.com/nhansen/SVanalyzer).

For the query callset, we used a VCF generated using either PacBio Revio HiFi reads (CHM13 hap.py analysis, Fig. 4) or Illumina HiSeq X PCR-Free reads (exploratory analysis, Fig. 5), both called with DeepVariant.

We ran the benchmark using hap.py as follows:

hap.py –engine vcfeval –stratifications <path/to/strats> –f <path/to/confident_regions.bed > -o <path/to/output > <path/to/bench.vcf > <path/to/query.vcf>

To generate the benchmarking plots, we used the stratified counts for true positive (TP), false positive (FP), and false negative (FN) from the *_extended.csv output file to calculate precision and recall as follows:

*Precision = QUERY.TP / (QUERY.TP + QUERY.FP)*
*Recall = TRUTH.TP / (TRUTH.TP + TRUTH.FN)*

The error bars were approximated using the binconf function from the Hmisc package in R using alpha = 0.05 and the wilson method, where "successes" were defined as QUERY.TP or TRUTH.TP and total observations was QUERY.TP + QUERY.FP or TRUTH.TP + TRUTH.FN for precision or recall, respectively.

A list of all files used for inputs to hap.py can be found in Supplementary Table 1

### Stratification coverage plots

To generate intra-chromosomal coverage plots as depicted in Supplementary Fig. 1, we divided each chromosome into 1Mbp windows, and then computed the number of bases within each window that fell within a given stratification BED file. Gaps were not considered in the case of non-T2T references.

The code to generate this can be found in the reference-agnostic snakemake pipeline (see below) at /workflow/scripts/python/bedtools/postprocess

/get_coverage_table.py

Note that we only showed several plots in this manuscript to highlight key findings; however, the snakemake pipeline (mentioned above) generates these plots for every single stratification bed file as part of its quality control output. These can be found for stratifications version 3.5 here: https://ftp-trace.ncbi.nlm.nih.gov/ReferenceSamples/giab/release/genome-stratifications/v3.5/validation/window_coverages/

### Exploring variant distributions in tandem repeats

The Q100 HG002 VCF (described above) was converted into a BED-like format where the chromosome coordinates were defined using the POS column and the length of the REF field. The length of the variant (length(ALT) - length(REF)) was also stored. The length was used to subset the variants into SNVs, INDELs (1-2 bp long or 3-49 bp long), and SVs (>49 bp). This BED-like file was then intersected (left outer join) with the v3.1 Tandem Repeats/Homopolymers stratification BED file from dipcall. We ignored multiallelic variants as well as variants that were partially outside a repeat region to simplify analysis.

### Exploring read coverage of each variant

We considered variants called with DeepVariant for the HG002 sample available at https://storage.googleapis.com/brain-genomics-public/research/sequencing/grch38/vcf/hiseqx/wgs_pcr_free/40x/HG002.hiseqx.pcr-free.40x.deepvariant-v1.0.grch38.vcf.gz for this exploratory analysis. For the extraction of the coverage, the mosdepth package v0.3.2 was used with the flag of setting the bin size to 1 based on the 40x PCR-free HiSeq X BAM file at https://storage.googleapis.com/brain-genomics-public/research/sequencing/grch38/bam/hiseqx/wgs_pcr_free/40x/HG002.hiseqx.pcr-free.40x.dedup.grch38.bam[45].

### Exploring distribution of genomic distance between consecutive variants

The genomic positions of variants for the sample GIAB HG002 were extracted from the HG002 DeepVariant VCF (described above). The distance between two consecutive variants were calculated. Using the matplotlib package v3.4.3 of Python v3.9, the histogram figure was depicted. We should also note that a portion of the genome is unknown (existing as Ns in the reference file), so no variant can be found in these regions. To make sure this fact does not influence our analysis, we discarded those variants neighboring unknown sequences in the reference genome which accounts for 328 variants out of 5,596,945 variants.

### Reporting summary

Further information on research design is available in the Nature Portfolio Reporting Summary linked to this article.

## Data availability

All versions of the genome stratifications up to v3.5 (the latest as of this writing) are available on an FTP site hosted by NCBI here at https://ftp-trace.ncbi.nlm.nih.gov/ReferenceSamples/giab/release/genome-stratifications/.

## Code availability

The initial work for this study (which originally took place at a hackathon) is freely available https://github.com/collaborativebioinformatics/NIST-GREX. The preliminary version of the code to generate stratifications is available at https://github.com/genome-in-a-bottle/genome-stratifications. The full pipeline in snakemake is available at https://github.com/usnistgov/giab-stratifications. A copy of the GitHub repository and HTML output of the snakemake pipeline are archived at Zenodo at https://zenodo.org/records/11176260.

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

## Acknowledgements

We thank Sierra Miller and Katherine Gettings for their feedback. Certain commercial equipment, instruments, or materials are identified to specify adequately experimental conditions or reported results. Such identification does not imply recommendation or endorsement by the National Institute of Standards and Technology, nor does it imply that the equipment, instruments, or materials identified are necessarily the best available for the purpose.

## Author contributions

N.D., F.J.S, J.W., S.M., and J.M.Z designed the study. N.D. implemented the pipeline. N.D., D.K., J.M., N.D.O, P.S, P.A.J, B.J., E.H., R.M. and S.M. performed the analyses. N.D., B.B., F.J.S, S.M., and J.M.Z organized the study. All authors reviewed and approved the manuscript.

## Competing interests

F.J.S. receives research support from Genetech, Illumina, ONT and Pacbio. B.B. is a full-time employee of DNAnexus. The remaining authors declare no competing interests
