## [Transparent Peer Review file · Nature Communications]

The GIAB genomic stratifications resource for human reference genomes

Corresponding Author: Dr Sina Majidian

Version 0:

Reviewer comments:

Reviewer #1

(Remarks to the Author)

Minor comments

-- Table 1. Colors are not appropriate for Tables. Maybe there is another way to communicate

-- It would be better to have scientific notations instead of 1e6 for all figures

-- CDS length figure 1 – would be nice to specify units? Bp?

-- Figure 2 . I am confused by the title of y axis. Low mapability length. Do the authors mean the length of each such region?

-- The formatting of Figure 3 can be improved as low categories is hard to see. One can consider making the break in the figure to how low categories better or use log scale for Y axis

-- The method section should be hap.py instead of happy

Major comments

1 – I was confused about the role of HPRC data. Is this part of GIAB? If not was it publicly downloaded? Did HPRC contain the same sample as GIAB (HG002). More details are needed. How many samples from HPRC were considered? What data was considered only Illumina or long reads as well?

2 – Section Benchmarking new stratification says “The draft benchmark”. I was not sure why this was a draft

3 – “or we can explain differences”. Not sure what it means? 4 – In terms of medically relevant regions. Is this information novel? Was it already reported in T2T paper?

5 – Figure 4 mentioned SV. I was wondering what types of SV were included (deletions, insertions?) and how they were inferred

6 – How HPRC was used? Was the assembly only used? Did authors use raw data from HPRC which can be superior to GIAB as it was a later technology? It would be nice to comment the limitations of GIAB for the task outlined in this project. Will the project benefit from the latest sequencing as done in HPRC project

7 – It would be interesting to speculate in the discussion section why the medically relevant regions were not significantly updated in the new genome releases

Reviewer #2

(Remarks to the Author)

The paper describes genomic stratification resource for human genomes that are useful for benchmarking and developing

variant callers.

The GIAB has been developing genomic stratifications and corresponding high-quality variant call sets for human genomes for several years that have proven to be immensely useful for the community. The paper primarily focuses on extension of the GIAB resource for the CHM13 reference which has become available recently. While the abstract of the paper seemed promising, most of the results presented in the paper seemed rather straightforward and it was not clear how these stratifications (or the proposed new stratifications) are useful for variant calling benchmarking and development of methods.

1. The section "Extending CHM13 stratifications" compares the statistics of coding sequences, low-mappability sequences and low GC content sequences across three different human assemblies. One notable finding was that certain chromosomes in the CHM13 assembly had additional low mappability sequences due to addition of repeat sequences that were missing in the previous human assemblies. There were several papers published for the CHM13 assembly and if I am not mistaken, a similar analysis was previously reported.

2. In Table 2, the length of three complex regions of the human genome are reported for the three assemblies and are quite similar. Again, the relevance of this analysis for genomic stratifications or variant calling is not clear. The new CHM13 assembly includes a lot of additional DNA sequence for segmental duplications that are missing in previous assemblies. The section on benchmarking new stratifications indicated that variant calling accuracy in segmental duplications was lower than in previous assemblies. It would be useful to see some more analysis of how these additional duplications in the CHM13 assembly impact the stratifications or the accuracy of variant calling (e.g. in coding regions that overlap segmental duplications).

3. The section on "exploring new features for future stratifications" analyzed variant complexity, inter-variant distances and read coverage. Although this was a "exploration of new features", without an analysis of how these variant stratifications impact variant calling accuracy (e.g. for short reads vs long reads), the section seems incomplete. It is not clear how the reader is supposed to interpret the analyses. The analysis of read coverage for variants showed that the average coverage was much lower for deletions compared to SNVs (38.2 vs 58.8). This is expected for short reads since most indels are in homopolymer runs and reads that don't cover the homopolymer tract presumably don't count towards the read coverage. Without further analysis (e.g. to show that higher read coverage is needed for calling deletions), the value of this analysis is not obvious.

4. Previous genomic stratifications from the GIAB have consisted of bed files with list of regions that can be called with high confidence along with a set of variant calls. Therefore, I would assume that the genomic stratification is dependent on the specific genome in addition to the reference genome assembly. For example, if a genome has an extra duplication of a large segment, then variant calling in this segment would be difficult.

Reviewer #3

(Remarks to the Author)

The manuscript "The GIAB genomic stratifications resource for human reference Genomes" prepared by Dwarshuis et. al. is an extension of the existing genome stratification resource to the CHM13-T2T reference. Stratifying the reference genome into different strata by their sequence properties such as complexity level, GC content, mappability, etc seems to be a sensible way for improve whole genome variant analysis. To extend the stratification to the CHM13-T2T reference is therefore a logical extension and I want to thank the authors for making this valuable resource.

My most important question after reading the manuscript is how I can apply this resource to my analysis to improve the results I can get from current bioinformatics software. Should I adjust my parameters for variant calling to accommodate the properties of the different strata? Or use different post-filtering criteria for variants belonging to different strata? I think the manuscript would have a much bigger impact by providing guidelines and use cases/examples on how to adjust our variant analysis in light of this stratification information. And examples may not be limited to variant calling. Other types of analysis such as haplotype phasing (with the new stratification class of "genomic distance between consecutive variants"), GWAS, etc as the authors suggested in their discussion would make the manuscript and the resource more impactful.

To convince the readers of the advantage of using stratifications, it would be nice if the authors can provide some specific examples after employing the stratification information versus without? This can be done on the T2T-CHM13 genome but not necessarily limiting to T2T as many readers (including myself) may not be aware of the resource before.

Pertaining to the T2T-CHM13 reference, it is of interest to know what or how much more can be gained using this new reference now with the stratifications? Are there any parts of the genome (which and where) where we can confidently call variants (especially with the help of stratification)?

On the manuscript level, most of the figures were made to deliver a comparison of the lengths or length ratios of the different strata by chromosome in the three references. I can see the exact region coordinates are provided to the public in BED format. I would appreciate if some kind of visualization can be provided to compare the locations of different strata in the three references, perhaps highlighting the differences?

For mappability calculation, I don't quite understand the strategy of identifying them and why a more stringent mapping is required for very difficult-to-map regions – why are you looking at variants/errors between 2 100/250bp regions? Are these paired regions? And slide-windows? Some explanation for the rationale of this algorithm in the supplementary would be appreciated. And again, how users should differentially apply these 2 unmappable strata in their analysis?

There is an error in the link to the ftp site (<https://ftp://ftp-trace.ncbi.nlm.nih.gov/ReferenceSamples/giab/release/genome-stratifications/>) on the github repo the authors should fix (there is a redundant "ftp")

Version 1:

Reviewer comments:

Reviewer #1

(Remarks to the Author)

All my comments were addressed

Reviewer #2

(Remarks to the Author)

My main concern in the previous review was how the stratifications are useful for variant calling benchmarking. This has been addressed to some extent in the revision through the analysis presented in Figure 4. My other main concern was the lack of clarity about the value of the more complete CHM13 reference assembly for genome stratification and variant calling. I think that similar concerns were also raised by another reviewer. Additional comments on the revised manuscript are as follows:

Response to question 1: see Supp Figure S5 and S6 from Aganezov et al. ref #48 (unique k-mer maps of chromosomes comparing hg38 and chm13). These figures represent very similar analysis as in the paper (Figure 2).

Figure 4b should be syntenic and non-syntenic rather than all/non-syntenic. This will make it easier to compare the two sets of regions.

The description of Figure 4 in the text can be improved with additional numbers/statistics.

Some comments on how the proposed stratifications are different from previous ones from the GIAB (<https://github.com/genome-in-a-bottle/genome-stratifications>) will be useful.

Out of curiosity, can the proposed pipeline (<https://github.com/ndwarshuis/giab-strats-smk>) also work for generating stratifications for any genome, in particular non-human genomes? This will make it more useful.

Reviewer #3

(Remarks to the Author)

I appreciate the authors' response to my comments. However, I still cannot grasp how the stratifications have an enormous impact on variant calling, other than informing researchers there are certain regions they would better skip or not trust (like HPs). But can this kind of filtering/screening already be achieved by using the score of SNV/indel calling from existing software? The added analysis of using different basecaller + variant calling of ONT sequences is likely helpful for software development, but does not demonstrate the utility of the resource to users in general.

So apparently the T2T-CHM13 reference does not offer much more in terms of variant calling, as the added ("non-syntenic") regions are most likely unmappable, and stratification is not able to find some sub-regions in these difficult regions where variant calling is feasible?

For the added supplementary figs 1 and 2, I wonder if by overlapping the GRCh38 and T2T-CHM13 low-mappability or GC-rich regions can identify some regions that have changes - from low to high mappability or from high to low GC-rich regions and thus lead to a change in variant calling ability? That is my intention of asking for a comparison in the profiles of the references.

Version 2:

Reviewer comments:

Reviewer #2

(Remarks to the Author)

The authors have substantially revised the Discussion and some of the Figures. I have no further comments.

Reviewer #3

(Remarks to the Author)

All my comments have been addressed.

Response to Reviewers Comments

We appreciate the helpful feedback and comments from three reviewers which provide us the chance to improve our work. We have addressed all the points which are detailed below.

Reviewer #1:

Minor comments

-- Table 1. Colors are not appropriate for Tables. Maybe there is another way to communicate

RESPONSE: We have updated the table by adding a column differentiating established stratification set and newly-developed exploratory stratification for future versions.

-- It would be better to have scientific notations instead of 1e6 for all figures

-- CDS length figure 1 – would be nice to specify units? Bp?

RESPONSE: Fixed.

-- Figure 2 . I am confused by the title of y axis. Low mapability length. Do the authors mean the length of each such region?

RESPONSE: Yes. We described it as “Total length of low-mappable regions” in the caption.

-- The formatting of Figure 3 can be improved as low categories is hard to see. One can consider making the break in the figure to how low categories better or use log scale for Y axis

RESPONSE: Thanks for the suggestion. We have updated Figure 3 (c-d) and made the y-axis in log scale.

-- The method section should be hap.py instead of happy

RESPONSE: Fixed.

Major comments

1 – I was confused about the role of HPRC data. Is this part of GIAB? If not was it publicly downloaded? Did HPRC contain the same sample as GIAB (HG002). More details are needed. How many samples from HPRC were considered? What data was considered only Illumina or long reads as well?

RESPONSE: To clarify this, we no longer use HPRC data in this manuscript so this is no longer relevant.

2 – Section Benchmarking new stratification says “The draft benchmark”. I was not sure why this was a draft

RESPONSE: The assembly-based HG002 benchmark is draft in the sense that it has not been published or officially released by Genome in a Bottle as of yet. However, it has been

characterized enough to be used in demonstrations such as this and has the important advantage (relative to the current GIAB v4.2.1 small variant benchmark) of including many of the “difficult” regions which we wanted to highlight with the stratifications in this publication. We have added a description why it is called “draft” in the Method section: “Note that we call these “draft” variants since this benchmark has not been officially evaluated and released” Additionally, we updated the Method section to include much more detail on how this assembly-based benchmark was generated, including a link to the public repository encoding the pipeline.

3 – “or we can explain differences”. Not sure what it means?

RESPONSE: We have completely revised the section, so this phrase is no longer present.

4 – In terms of medically relevant regions. Is this information novel? Was it already reported in T2T paper?

RESPONSE: The only reference to medically relevant regions here are the VDJ, KIR, and MHC loci. These are well-known and well-characterized immunological regions and all three are included in GRCh37 and GRCh38 (ie not novel). These are included as stratifications due to their polymorphic nature (as explained in the paper, hopefully better in this revision).

5 – Figure 4 mentioned SV. I was wondering what types of SV were included (deletions, insertions?) and how they were inferred

RESPONSE: For this figure, the variant types are simply classified by length, where SVs are >50bp long. Both insertions and deletions were included in the “SV” label. This was derived from the VCF file itself using the difference b/t the REF and ALT columns (note that this was erroneously stated to be a small variant benchmark in the previous version of the manuscript which would obviously not provide large SVs; we corrected this in the updated manuscript version).

6 – How HPRC was used? Was the assembly only used? Did authors use raw data from HPRC which can be superior to GIAB as it was a later technology?

RESPONSE: We no longer use HPRC at all in this manuscript revision. Instead we use truth variant callsets derived from the HG002 Q100 T2T assembly as explained further below and in the updated method section. The Q100-based callsets should be superior to both GIAB’s v4.2.1 benchmarks and any callsets derived from the HPRC assemblies (see next).

It would be nice to comment the limitations of GIAB for the task outlined in this project. Will the project benefit from the latest sequencing as done in HPRC project

RESPONSE: This is indeed true for the current GIAB v4.2.1 small variant benchmark. In this updated manuscript version we no longer use these benchmarks (with one exception,

described below). Instead, we used an assembly-based benchmark, which is based on assembly-assembly alignment vs read-assembly alignment (described in more detail in the methods section). Specifically, we used the Q100 HG002 T2T assembly to make these benchmarks, which have advantages beyond GIAB v4.2.1 and HPRC assemblies due to a) the newer technology used to make the assemblies, including the extensive polishing and b) the fact that these assemblies were derived using haplotype-reference alignments vs read-haplotype alignments, which by default will provide more structural information and hence greater accuracy.

Further note that we still used the v4.2.1 benchmark in the case of comparing ONT performance across different platform iterations (see Fig 4c). This analysis was not in the previous version of the manuscript, and the reason we didn't use the HG002 benchmark is because the ONT data available for this analysis was only provided for HG003, and we have yet to build an assembly-based benchmark for HG003. This should not affect the final conclusion however, as the point of that specific experiment was to demonstrate how the stratifications can be used to assess performance improvements for a specific platform, and not to investigate how the additional "difficult" regions in CHM13 affect benchmarking, which indeed would be better served with a more-complete benchmark (exactly as is done in Fig 4a-b)

7 – It would be interesting to speculate in the discussion section why the medically relevant regions were not significantly updated in the new genome releases

RESPONSE: A deep analysis of new medically relevant regions in the CHM13 is beyond the scope of this study, because GIAB's stratifications are primarily focused on regions that are technically challenging for variant calling in the GIAB samples, which generally do not include medically relevant variants. Discovering new medically relevant regions in CHM13 will likely require using this reference with large cohorts that have medical indications, and cannot generally be discovered from GIAB's normal samples.

Reviewer #2:

The paper describes genomic stratification resource for human genomes that are useful for benchmarking and developing variant callers. The GIAB has been developing genomic stratifications and corresponding high-quality variant call sets for human genomes for several years that have proven to be immensely useful for the community. The paper primarily focuses on extension of the GIAB resource for the CHM13 reference which has become available recently. While the abstract of the paper seemed promising, most of the results presented in the paper seemed rather straightforward and it was not clear how these stratifications (or the proposed new stratifications) are useful for variant calling benchmarking and development of methods.

RESPONSE: We are thankful for your positive evaluation of the GIAB resources. Admittedly the usefulness was not argued or demonstrated very well in the first revision. We have since addressed this in several ways.

First, we added several benchmarking analyses using the new CHM13 stratifications. While we performed one benchmarking analysis in the previous revision, the results were buried in the supplemental data. These updated results are presented in what is now Figure 4, and includes a demonstration of how the addition of more difficult regions in CHM13 generally decreases precision and recall relative to GRCh37/38. Importantly, this analysis was performed with a benchmark derived from the Q100 T2T HG002 assembly, which is important since (relative to current GIAB benchmarks) this includes variants in these difficult regions and thus allows CHM13 to be fully tested. Our analysis also included an experiment wherein we subsetted to the non-syntenic regions between CHM13 and GRCh38 (i.e., new regions in CHM13) and showed how the scores drop dramatically relative to non-subsetted regions. All of this demonstrates how the new CHM13 reference itself is useful, and also shows how the CHM13 stratifications that we developed are useful as these performance differences are not evenly distributed.

Second, we demonstrated how the stratifications can provide nuanced performance assessments for different technology iterations (Fig 4b); in our case we used ONT's base and variant callers as an example. Briefly, we showed how ONT's updated software generally improves variant calling precision/recall, but to a lesser degree in homopolymers. While this is only one example, one could easily extrapolate this to analogous situations where one wants to assess progress when developing any sequencing hardware or software. Additionally, one can imagine how this is useful in comparing different platforms with different cost-benefit functions.

Thirdly, we have assessed the precision and recall of the HG002 Illumina HiSeq X DeepVariant callset (as the query) with the Q100 T2T HG002 as (as the truth) using the three newly designed stratifications, namely, variant complexity in tandem repeats, distribution of genomic distance between consecutive variants, and read coverage of each variant. The results are depicted in Figure 5f-h.

1. The section "Extending CHM13 stratifications" compares the statistics of coding sequences, low-mappability sequences and low GC content sequences across three different human assemblies. One notable finding was that certain chromosomes in the CHM13 assembly had additional low mappability sequences due to addition of repeat sequences that were missing in the previous human assemblies. There were several papers published for the CHM13 assembly and if I am not mistaken, a similar analysis was previously reported.

RESPONSE: We did complete research about this (e.g. publications listed below) and we couldn't find similar analysis in the literature.

- "The complete sequence of a human genome" where genome annotation is detailed.
- "Epigenetic patterns in a complete human genome" and "From telomere to telomere: The transcriptional and epigenetic state of human repeat elements" describing the epigenetic annotation.
- "excluderanges: exclusion sets for T2T-CHM13, GRCh38, and other genome assemblies" including gaps, other regions found using ENCODE blacklist software should be excluded when analysing ChIP-seq data.
- "Segmental duplications and their variation in a complete human genome" which identifies duplication regions.

We would highlight that the CHM13 stratification resource has not been reported in the previous T2T publications and is part of the innovation of this manuscript.

2. In Table 2, the length of three complex regions of the human genome are reported for the three assemblies and are quite similar. Again, the relevance of this analysis for genomic stratifications or variant calling is not clear. The new CHM13 assembly includes a lot of additional DNA sequence for segmental duplications that are missing in previous assemblies. The section on benchmarking new stratifications indicated that variant calling accuracy in segmental duplications was lower than in previous assemblies. It would be useful to see some more analysis of how these additional duplications in the CHM13 assembly impact the stratifications or the accuracy of variant calling (e.g. in coding regions that overlap segmental duplications).

RESPONSE: We appreciate the reviewers comments on describing the importance and relevance of the analysis. As mentioned earlier, we addressed these shortcomings of the benchmarking section by including an analysis wherein we subsetted the non-syntenic regions between GRCh38 and CHM13, and showed that the precision and recall are much lower (see Figure 4).

3. The section on "exploring new features for future stratifications" analyzed variant complexity, inter-variant distances and read coverage. Although this was a "exploration of new features", without an analysis of how these variant stratifications impact variant calling accuracy (e.g. for short reads vs long reads), the section seems incomplete. It is not clear how the reader is supposed to interpret the analyses. The analysis of read coverage for variants showed that the average coverage was much lower for deletions compared to

SNVs (38.2 vs 58.8). This is expected for short reads since most indels are in homopolymer runs and reads that don't cover the homopolymer tract presumably don't count towards the read coverage. Without further analysis (e.g. to show that higher read coverage is needed for calling deletions), the value of this analysis is not obvious.

RESPONSE: Based on the suggestions of the reviewer, we have completely revised the section "Exploring new features for future stratifications" to demonstrate how these are useful. We investigated the impact of the three new stratifications (sequencing coverage, genomics distance between variants and number variants in a tandem repeat) on precision and recall of variant calling. (Please see Figure 5f-h). Specifically we found that more variants in a tandem repeat region corresponds to lower precision and recall in variant calling. Also, regarding the new stratification of genomic distance between consecutive variants, the quality of called variants is decreased when variants are close to each other. The analysis on coverage values showed that precision and recall of variant calling the highest when the variant in a region with mean of coverage 40X (More details in the main manuscript in the section "Exploring new features for future stratifications"). Each of these stratifications were tested using hap.py and the Q100 HG002 benchmark as also done in Figure 4.

4. Previous genomic stratifications from the GIAB have consisted of bed files with list of regions that can be called with high confidence along with a set of variant calls. Therefore, I would assume that the genomic stratification is dependent on the specific genome in addition to the reference genome assembly. For example, if a genome has an extra duplication of a large segment, then variant calling in this segment would be difficult.

This is somewhat confusing and we tried to clarify this in our methods. Briefly, there are several sets of bed files that are being conflated here:

1. Benchmark (or confident) regions: the GIAB small variant benchmarks consist of a VCF file and a bed file denoting where the variants in the VCF are "reliable" for a particular sample. We updated the manuscript to use assembly-based benchmarks rather than the current v4.2.1 small variant benchmark (based on read alignments) as these are more accurate in the difficult regions we wished to analyze. This benchmark contains an analogous bed file which more-or-less depicts where the variant calls are "reliable." Each benchmark is genome-specific and called relative to a reference.
2. Stratification bed files: the stratifications (ie the main topic in this paper) are bed files which are reference-specific and based upon inherent properties of the reference for which distinct boundaries can be drawn (high GC, homopolymers, etc). These are not the same as confident regions described in (1). When benchmarking using a tool such as hap.py, the two callsets are subsetted to the confident regions, and if the stratifications are supplied, the resulting benchmarking metrics (precision, recall, etc) are further subsetted within each stratification. In general, these have no relation to any other haplotypes, with a few exceptions:
 - a. The stratifications for GRCh38 include genome-specific stratifications for each of the GIAB samples (HG001-HG007). These stratifications contain regions which correspond to known difficulties within these genomes, such

as complex structural variants, copy number variants, etc. While these do exist, we did not focus on them in this manuscript as we do not have the same data (currently) for these types of difficult regions relative to CHM13.

- b. In this manuscript we proposed several new stratifications which we might use. Notably, these all relied on HG002 variants relative to a reference, so these stratifications (if we end up including them in future releases) would also be genome-specific as described above.

Reviewer #3:

The manuscript “The GIAB genomic stratifications resource for human reference Genomes” prepared by Dwarshuis et. al. is an extension of the existing genome stratification resource to the CHM13-T2T reference. Stratifying the reference genome into different strata by their sequence properties such as complexity level, GC content, mappability, etc seems to be a sensible way for improve whole genome variant analysis. To extend the stratification to the CHM13-T2T reference is therefore a logical extension and I want to thank the authors for making this valuable resource.

RESPONSE: We appreciate your encouraging assessment of the GIAB stratification resource.

My most important question after reading the manuscript is how I can apply this resource to my analysis to improve the results I can get from current bioinformatics software. Should I adjust my parameters for variant calling to accommodate the properties of the different strata? Or use different post-filtering criteria for variants belonging to different strata? I think the manuscript would have a much bigger impact by providing guidelines and use cases/examples on how to adjust our variant analysis in light of this stratification information. And examples may not be limited to variant calling. Other types of analysis such as haplotype phasing (with the new stratification class of “genomic distance between consecutive variants”), GWAS, etc as the authors suggested in their discussion would make the manuscript and the resource more impactful.

RESPONSE: Thank you for the suggestion. While this was not intended to be a methods paper, we concede that the intended use-cases for the stratifications were difficult to understand. In this new version of the manuscript, we updated/added several analyses, namely analyzing two ONT-based callsets created with either guppy4+clair1 or guppy5+clair3, and studying impact of reference genome on variant calling performance, and how non-syntenic regions impact the variant quality. These analyses showed how stratifications could be useful in practice (described further below).

We also reference other manuscripts that rely on these stratifications to show their impact, such as this addition to the Discussion “For example, GIAB has used these stratifications to exclude problematic regions from benchmarks, such as long homopolymers in a new benchmark for chromosomes X and Y²⁹.”

To convince the readers of the advantage of using stratifications, it would be nice if the authors can provide some specific examples after employing the stratification information versus without? This can be done on the T2T-CHM13 genome but not necessarily limiting to T2T as many readers (including myself) may not be aware of the resource before.

RESPONSE: We added a much more comprehensive benchmarking analysis with these new stratifications in Fig 4. To directly address the stratification vs no-stratification contrast, the most direct result we can highlight is Fig4c where we compare different

version of ONT's analysis pipeline. The "AllAutosomes" corresponds to "the entire genome" (since our benchmark here didn't include X or Y). Stratifying by homopolymers/tandem repeats or the complement shows dramatic differences relative to "AllAutosomes," and such an analysis would be very difficult without stratifications for these regions. To the point of CHM13, this analysis used GRCh38 since we only had VCF files relative to this reference.

Pertaining to the T2T-CHM13 reference, it is of interest to know what or how much more can be gained using this new reference now with the stratifications? Are there any parts of the genome (which and where) where we can confidently call variants (especially with the help of stratification)?

RESPONSE: This is also addressed in our updated Fig 4, specifically Fig4a-b. This shows how the precision and recall generally decrease with each reference, where CHM13 is the lowest and GRCh37 is the highest. This corresponds with how complete each reference is (i.e. CHM13 is more complete than GRCh38 which is more complete than GRCh37). The stratifications show how these performance differences are not evenly distributed, both within and between references. Furthermore, in Fig4b we show how the new regions in CHM13 relative to GRCh38 (non-syntenic) are much more difficult, as reflected by the lower scores.

To directly answer the question, what can be gained is a more realistic sense of difficulty throughout the genome, since the scores in an incomplete reference are higher (i.e. overinflated) relative to a complete T2T reference (CHM13).

On the manuscript level, most of the figures were made to deliver a comparison of the lengths or length ratios of the different strata by chromosome in the three references. I can see the exact region coordinates are provided to the public in BED format. I would appreciate if some kind of visualization can be provided to compare the locations of different strata in the three references, perhaps highlighting the differences?

RESPONSE: This was a helpful suggestion. We created "intra-chromosomal" plots which show the coverage of a given BED file for 1Mbp windows. We only chose to show two in the paper which corresponded to the specific analyses we wished to highlight (Supplemental Figures 1 and 2) since the plots ended up being rather big; however, these will be available for every single stratification BED file in the next stratification release since this is an informative quality control visualization we had not previously created (and this is all stated in the new manuscript revision).

For mappability calculation, I don't quite understand the strategy of identifying them and why a more stringent mapping is required for very difficult-to-map regions – why are you looking at variants/errors between 2 100/250bp regions? Are these paired regions? And slide-windows? Some explanation for the rational of this algorithm in the supplementary

would be appreciated. And again, how users should differentially apply these 2 unmappable strata in their analysis?

RESPONSE: We had included an explanation of this but it was in the supplemental. We added further rationale for this in the corresponding results section where we discuss the addition of hard-to-map regions: "However, for very difficult-to-map regions we permitted no mismatches or INDELS between each 250 bp region and any other region. Note that 100 and 250 bp correspond approximately to two common lengths used for short-read sequencing, so these regions can be interpreted as those that are difficult to map for short reads."

There is an error in the link to the ftp site (<https://ftp//ftp-trace.ncbi.nlm.nih.gov/ReferenceSamples/giab/release/genome-stratifications/>) on the github repo the authors should fix (there is a redundant "ftp")

RESPONSE: Thank you for pointing this out. The link is fixed.

Response to Reviewers Comments

We thank the reviewers for their helpful comments. We have addressed all the points and answered all the comments below.

Reviewer #1

All my comments were addressed.

Reviewer #2

My main concern in the previous review was how the stratifications are useful for variant calling benchmarking. This has been addressed to some extent in the revision through the analysis presented in Figure 4.

RESPONSE: Besides the analysis shown in the revised Figure 4, which clearly highlights the utility of the GIAB stratifications resource for variant assessment and benchmarking of different variant calling methodologies, a substantial benefit is having this newly centralized and harmonized resource. This resource can be utilized for different applications such as for variant filtering (e.g., in low mappable regions), to assess sequencing biases (GC biases) and for variant representations impact (e.g., in tandem repeat and homopolymers) to name a few examples. Furthermore this harmonization across the two GRCh37/38 and CHM13 reference genomes will avoid confusion and inconsistencies in future studies of variant calling or benchmarking and will remain a long term resource for GIAB as well as other important benchmarking studies (e.g. FDA-SEQC2).

As described in the current manuscript, another specific example of stratification use case is to stratify the performance of variant callers to know which tools perform better in a genomic region with specific profile. We previously implemented this in a limited scenario for GRCh38, published in PrecisionFDA Truth Challenge V2. Please see Figure 5

<https://www.sciencedirect.com/science/article/pii/S2666979X22000581>.

In addition to benefits for variant calling analysis, we could also mention benefits of the genomic stratifications in other applications. The stratifications have also applied for characterization of assembly accuracy (e.g., in <https://www.nature.com/articles/s41586-022-05325-5>). Another application could be the error rate of RNA/DNA sequencing instruments. Depending on the genomic context, some sequencing devices might have a higher error rate. This will be a new line of research to systematically investigate the potential sequencing biases in different low-mappability, repeat regions or regions with extreme GC content (<https://doi.org/10.1093/nar/gks001>, <https://doi.org/10.1093/nar/qkr425>) for either short or long read.

We believe that the GIAB stratification resource will be widely used by the scientific community. To fully address the reviewer's concern, we have completely revised the section of **Discussion & Conclusion** to highlight all the benefits of the resource.

My other main concern was the lack of clarity about the value of the more complete CHM13 reference assembly for genome stratification and variant calling. I think that similar concerns were also raised by another reviewer.

RESPONSE: To address the concerns of both reviewers, we improved the manuscript to more clearly show the benefits of stratification sets for CHM13. Specifically, here is a summary of what we changed:

- We added Figure 4d which shows a comparison between HiFi and Illumina callsets using CHM13 as the reference and the HG002 Q100 benchmark as the truth. The most extreme differences on this plot were for segmental duplications and mappability recall (dramatically in favor of HiFi reads). HiFi did not perform perfectly on recall either, showing that many of these regions are still difficult even with long reads. CHM13 includes many of these difficult regions which were not present in GRCh38, which simultaneously raises the bar for what a sequencing platform needs to accomplish in order to “find all variants in the genome” and also (along with our stratifications) provides a measuring stick by which to assess progress toward that goal.
- We added Supplementary Figure 2, which shows the number of variants (from the Q100 HG002 benchmark) that are in the non-syntenic regions between CHM13 and GRCh38. From this we can see that there are 1000s of variants in CHM13 that are not in GRCh38, thus using CHM13 dramatically lowers the difficulty of studying these variants (many of which likely have biological significance).

We added paragraphs starting from lines 366 and 553 to the Results and Discussion respectively to address the comment (please see the marked-up manuscript file with tracked changes and line numbers).

Supplementary Figure 3: Comparison of variants in syntenic and non-syntenic regions. SV = structural variant (variant 50 bp or longer).

Additional comments on the revised manuscript are as follows:

Response to question 1: see Supp Figure S5 and S6 from Aganezov et al. ref #48 (unique k-mer maps of chromosomes comparing hg38 and chm13). These figures represent very similar analysis as in the paper (Figure 2).

RESPONSE: These are indeed similar in that those figures broadly show an analysis of “mappability” across GRCh38 and CHM13. However, these are not as immediately actionable as what we created with our mappability stratifications. First, the stratifications are only for 2 lengths, which correspond to 2 common lengths for short reads. This information is almost impossible to derive from the figure due to the scaling of the heatmap, and it would be less precise regardless. Second, “mappability” for our stratifications is in terms of alignment parameters that allow for a small degree of mismatch. This is more realistic considering that most will be aligning short reads using an aligner rather than exact matching (which is what the figure used).

Independent of this, our stratifications also include GRCh37, which is still used by many clinical laboratories and thus allows these users to benefit from this work and understand how transitioning to newer references could improve accuracy in certain regions.

Figure 4b should be syntenic and non-syntenic rather than all/non-syntenic. This will make it easier to compare the two sets of regions.

RESPONSE: This is indeed a more sensible way to compare the two. We updated the script to reflect this; however, the graph in 4B barely changed with the syntenic numbers increasing marginally. Note that the colors also flipped.

B.

The description of Figure 4 in the text can be improved with additional numbers/statistics.

RESPONSE: We added numbers to the figure itself corresponding to the value of each bar. We thought this was the most concise way to present these numbers precisely (see updated figure in previous comment).

We also added a brief explanation of what the error bars mean in the bar graph (“In this figure, the value of each bar is the aggregated phred-scaled score, and the error bars are the estimated 95% binomial confidence intervals.” line 293)

Some comments on how the proposed stratifications are different from previous ones from the GIAB (<https://github.com/genome-in-a-bottle/genome-stratifications>) will be useful.

RESPONSE: We added a new paragraph (starting from line 646) in the discussion to address this.

Out of curiosity, can the proposed pipeline (<https://github.com/ndwarshuis/giab-strats-smk>) also work for generating stratifications for any genome, in particular non-human genomes? This will make it more useful.

RESPONSE: Thank you for the useful question.

The pipeline may (in theory) be used for any haploid or diploid genome depending on the data one has available. Some stratifications can be derived directly from the reference fasta itself, and others require externally-generated annotations (like the output from TandemRepeatFinder for example). In the particular case of the XY chromosomes, the pipeline currently can only handle standard references that include the X and Y chromosomes, since Y is hardcoded. Future developments are aimed at enabling generation of stratifications for diploid reference assemblies, including those with an XX karyotype. XY stratifications also assume that categories such as “pseudoautosomal regions”, “X-transposed” and “ampliconic” are sensible for the genome under consideration as these are also hardcoded.

We added a discussion point for this (starting from line 663) as well as a table showing which stratifications require external data sources. We also added the hardware/software requirements to make this easier.

(Remarks on code availability): The stratification files for the different genomes are available. The README file gives an appropriate description of the resource.

RESPONSE: Thanks.

Reviewer #3

I appreciate the authors' response to my comments. However, I still cannot grasp how the stratifications have an enormous impact on variant calling, other than informing researchers there are certain regions they would better skip or not trust (like HPs). But can this kind of filtering/screening already be achieved by using the score of SNV/indel calling from existing software?

RESPONSE: Assuming that the reviewer is referring to the variant quality score from the VCF file, this score is complementary to our stratifications for several reasons. First, the meaning of this score is dependent on what the developers decided to use as inputs. While they may take a limited amount of contextual information into account, it is generally far less than the contextual richness one could derive using the stratifications. Second, this one score is just a single number. This generally doesn't perfectly reflect the difficulty of calling a variant, and also it doesn't provide any additional information beyond "one can trust this with X confidence." Part of the value of having many different stratification BED files is that understanding where a variant is located can provide a framework from which one can create and test hypotheses. This is useful for anyone developing a platform (or software for said platform) because it can inform **why** a given call might be inaccurate. And for everyone else who is using said platforms and software, it can help the user make a better decision about what to choose to avoid errors in whatever context they desire. To address the reviewer's point for readers of the manuscript, we added this point in the Discussion section (starting in line 582 in the marked-up manuscript file with tracked changes).

The added analysis of using different basecaller + variant calling of ONT sequences is likely helpful for software development, but does not demonstrate the utility of the resource to users in general.

RESPONSE: We acknowledge that the software development community would be most immediately interested in this result at face-value. However, one can extrapolate from this figure and imagine that the exact same strategy may be used to compare "X vs Y" (where in this case these are two versions of ONT, they could as well be different mappers or platforms). Thus, this analysis is an example of a wide range of analyses that the resource could make possible. We described the importance of the resource in the Introduction section. We now updated the Discussion section (lines 575-587) to highlight other broad applications of the resource.

To make this point clearer, we added an analysis comparing Illumina and HiFi on CHM13 with the Q100 benchmark (Figure 4D). In this figure we show in which stratifications either platform excels. Depending on which stratifications one cares about, this will help users decide (where "users" in this case is anyone who potentially has a need for long and/or short read sequencing).

D.

So apparently the T2T-CHM13 reference does not offer much more in terms of variant calling, as the added ("non-syntenic") regions are most likely unmappable, and stratification is not able to find some sub-regions in these difficult regions where variant calling is feasible?

RESPONSE: For a region to be “unmappable” one needs to consider the sequencing platform and associated software. There are many regions that are new in CHM13 that are indeed unmappable for short reads. For long reads (depending on how long), many of these regions become mappable and therefore accessible for variant calling.

Even with long reads, there are additional challenges, however. Since the non-syntenic regions have many repetitive regions (centromeres, satellites, etc), variants may have many possible and valid representations, and there are no current standards for how to do this. Furthermore, there are still some regions (rDNA arrays for example) that are repetitive and large enough that mapping (and variant calling by extension) will remain difficult until technology improves. While this may limit CHM13’s full potential in the short term, the fact that these regions now have a complete reference will give sequencing technology developers a target, and once a given platform hits this target these regions will become accessible.

Hopefully this was at least partly addressed with the figure addition in the previous question, which quantified the exact difference in performance between HiFi and Illumina for difficult to map regions (and more) in CHM13 specifically. Notice in that figure that the recall in segmental duplications and low-mappability are not perfect even for HiFi reads, although they are much better than Illumina as expected. This shows that there is still room for improvement. We added this point in the discussion (starting in line 635)

Furthermore, the other advantage of using CHM13 is that it is complete in terms of its representation of the entire genome (whereas GRCh38 is only 92% complete). This means that any variants in the “dark” 8% of the genome are now accessible if one uses CHM13. We added an additional figure (Supplemental Figure 3; provided below) to show (using the Q100 HG002 benchmark) that one can expect to find on the order of 1000s of SNVs and 100s of INDELs in the non-syntenic regions, many of which might have biological significance and up

until recently have been very difficult to study. We also added a discussion point for this starting in line 589.

For the added supplementary figs 1 and 2, I wonder if by overlapping the GRCh38 and T2T-CHM13 low-mappability or GC-rich regions can identify some regions that have changes - from low to high mappability or from high to low GC-rich regions and thus lead to a change in variant calling ability? That is my intention of asking for a comparison in the profiles of the references.

RESPONSE: Thank you for the interesting question.

For mappability, moving to CHM13 from GRCh38 might expand/contract hard-to-map regions by several mechanisms:

1. GRCh38 has known segmental duplications that are either falsely duplicated or falsely collapsed relative to the general population (including CHM13). In the case of false duplications, the “correct” copy which is only present in CHM13 would likely be easier to map (and the reverse for false collapses). Many of these regions were previously explored as part of the T2T effort <https://www.science.org/doi/10.1126/science.abl3533>, and subsequent work showed

how GRCh38 can be improved to reduce errors <https://genomebiology.biomedcentral.com/articles/10.1186/s13059-023-02863-7>. Our stratifications include BED files for GRCh38 delineating the affected regions.

2. CHM13 adds many new regions that are simply harder to map due to their repetitiveness (centromeres, satellites, etc)
3. CHM13 adds additional copies of segmental duplications that exist in GRCh38; these copies are likely to be harder to map in CHM13 due to the additional copies.

We tried to explore this further by lifting the mappable regions from GRCh38 onto CHM13 and filtering out the regions which landed in the non-syntenic regions (which we deemed untrustworthy since these regions were previously determined to be missing from GRCh38 or challenging to liftover). We then intersected these with the mappable regions from CHM13 itself to get regions that were mappable in only one reference or both. The following figure shows the fraction of bases belonging to each group for 1Mb windows. While there are differences, the fact that most of these are either in both references or CHM13 means that much of this can be explained by the reduction in gaps between GRCh38 and CHM13 (point 2 above). There are some regions that are mappable only in GRCh38 but these are relatively few and many of them are concentrated around the centromere, which means they likely were mappable simply because much of the centromere was not included in GRCh38 (and this is not true in CHM13). There are likely other explanations (see points 1 and 3 above) for why these differences manifest, but they are also relatively few in number, and we provide stratifications for dealing with these already. Given that this was a liftover (which are inherently error prone, especially when dealing with references that have large degrees of difference between them), and previous work referenced above have partly addressed this question, we have decided that this analysis is outside of the scope of this paper.

For GC content, we are unsure how the reference would be likely to change from GC-rich to GC-poor, or vice versa. The GC rich/poor regions already in GRCh38 are also likely to still be in CHM13, which means that any additional GC rich/poor regions in CHM13 would simply coincide with the regions that are new to CHM13, such as large repeats in the centromere and heterochromatin.